# Pitfalls of DualTracer 99m-Technetium (Tc) Pertechnetate and Sestamibi Scintigraphy before Parathyroidectomy: Between Primary-Hyperparathyroidism-Associated Parathyroid Tumour and Ectopic Thyroid Tissue

**DOI:** 10.3390/medicina60010015

**Published:** 2023-12-21

**Authors:** Mara Carsote, Mihaela Stanciu, Florina Ligia Popa, Oana-Claudia Sima, Eugenia Petrova, Anca-Pati Cucu, Claudiu Nistor

**Affiliations:** 1Department of Endocrinology, Carol Davila University of Medicine and Pharmacy, 050474 Bucharest, Romania; carsote_m@hotmail.com (M.C.); jekined@yahoo.com (E.P.); 2Department of Clinical Endocrinology V, C.I. Parhon National Institute of Endocrinology, 020021 Bucharest, Romania; oanaclaudia1@yahoo.com; 3Department of Endocrinology, Faculty of Medicine, “Lucian Blaga” University of Sibiu, 550024 Sibiu, Romania; 4Department of Physical Medicine and Rehabilitation, Faculty of Medicine, “Lucian Blaga” University of Sibiu, 550024 Sibiu, Romania; 5PhD Doctoral School, Carol Davila University of Medicine and Pharmacy, 020021 Bucharest, Romania; ancutapati@gmail.com; 6Thoracic Surgery Department, Dr. Carol Davila Central Military Emergency University Hospital, 010242 Bucharest, Romania; ncd58@yahoo.com; 7Department 4—Cardio-Thoracic Pathology, Thoracic Surgery II Discipline, Carol Davila University of Medicine and Pharmacy, 050474 Bucharest, Romania

**Keywords:** thyroid, parathyroid, 99m-Technetium, scintigraphy, thyroidectomy, parathyroidectomy, ectopic, primary hyperparathyroidism

## Abstract

Diagnosis of primary hyperparathyroidism (PHP) is based on blood assessments in terms of synchronous high calcium and PTH (parathormone), but further management, particularly parathyroid surgery that provides the disease cure in 95–99% of cases, requires an adequate localisation of the parathyroid tumour/tumours as the originating source, with ultrasound and 99m-Technetium (99m-Tc) sestamibi scintigraphy being the most widely used. We aimed to introduce an adult female case diagnosed with PHP displaying unexpected intra-operatory findings (ectopic thyroid tissue) in relation to concordant pre-operatory imaging modalities (ultrasound + dual-phase 99m-Tc pertechnetate and sestamibi scintigraphy + computed tomography) that indicated bilateral inferior parathyroid tumours. A sudden drop in PTH following the removal of the first tumour was the clue for performing an extemporaneous exam for the second mass that turned out to be non-malignant ectopic thyroid tissue. We overviewed some major aspects starting from this case in point: the potential pitfalls of pre-operatory imaging in PHP; the concordance/discordance of pre-parathyroidectomy localisation modalities; the need of using an additional intra-operatory procedure; and the clues of providing a distinction between pathological parathyroids and thyroid tissue. This was a case of adult PHP, whereas triple localisation methods were used before parathyroidectomy, showing concordant results; however, the second parathyroid adenoma was a false positive image and an ectopic thyroid tissue was confirmed. The pre-operatory index of suspicion was non-existent in this patient. Hybrid imaging modalities are most probably required if both thyroid and parathyroid anomalies are suspected, but, essentially, awareness of the potential pitfalls is mandatory from the endocrine and surgical perspectives. Current gaps in imaging knowledge to guide us in this area are expected to be solved by the significant progress in functional imaging modalities. However, the act of surgery, including the decision of a PTH assay or extemporaneous exam (as seen in our case), represents the key to a successful removal procedure. Moreover, many parathyroid surgeons may currently perform 4-gland exploration routinely, precisely to avoid the shortcomings of preoperative localisation.

## 1. Introduction

Diagnosis of primary hyperparathyroidism (PHP) is based on blood assessments in terms of synchronous high calcium and PTH (parathormone), but further management, particularly parathyroid surgery that provides the disease cure in 95–99% of cases, requires an adequate localisation of the parathyroid tumour/tumours as the originating source of the hormonal and biochemistry anomalies [1,2].

Parathyroid imaging represents a mandatory pre-operatory step and a large spectrum of tools are currently available, starting with the mostly frequently used, namely cervical ultrasound. While echography has the advantage of being non-expensive and not representing a source of irradiation, other techniques use different radiation doses such as parathyroid scintigraphy and four-dimensional (4D) computed tomography (CT) scans. Also, an enhancement in the baseline features as provided by the neck ultrasound may be provided by 4D MRI (magnetic resonance imaging) [3,4,5].

99m-Technetium (99m-Tc) is used as a tracer in relation to the parathyroid uptake (regardless if ectopic or orthotopic) and 99m-Tc sestamibi is commonly applicable, for instance, as single-photon scintigraphy with 99mTc-sestamibi or a scintigraphy with a dual tracer (99m-Tc pertechnetate and 99m-Tc sestamibi), with either variant being associated or not with thyroid subtraction [6,7].

99m-Tc scintigraphy represents a key imaging tool to address the parathyroid glands and, in most countries, almost every patient that is referred to parathyroidectomy had underwent at least one imaging procedure before surgery, mostly ultrasound in addition to 99m-Tc sestamibi scintigraphy [8,9]. Moreover, the results may be enhanced by adding SPECT (single-photon-emission computed tomography) or SPECT/CT regardless of an orthotopic or ectopic parathyroid tumour [10,11]. The underlying pathway of this molecular imaging is represented by the detection of the mitochondrial content at the level of parathyroid tissue, noting that hyper-functional parathyroid glands in PHP are associated with a high-oxyphil-cell-related abundant mitochondrial count [12,13]. The molecular imaging approach brings more functional information when compared to structural (standard) techniques such as CT or MRI. Further on, PET/CT (positron emission tomography), for instance, with ^18^F-Fluorocholine or ^11^C-methionine and PET/MRI (which requires lower radiation dose than PET/CT or CT scan), provides useful dynamic captures [7,14,15]. Additionally, angiographic selective venous sampling represents an alternative to the multimodal diagnosis approaches in the parathyroid field, but its exact placement among other modalities currently represents an emergent issue to be solved [16,17]. Importantly, this is an invasive tool that is largely reserved for re-operative cases in which imaging has not identified a target.

However, overall, despite recent progress in this specific area, there is no standard (unanimous) consensus with concern to the optimal imaging diagnosis before parathyroidectomy, with the applied techniques largely depending on one centre experience, the availability/costs of these procedures and also an individualised approach in relation to the clinical circumstances of a patient.

Of note, the clinical decision involving both endocrinology and surgery levels remains the core of this multidisciplinary management. While a skilled surgeon relates to the optimum parathyroidectomy outcome with regard to PHP and secondary (tertiary) hyperparathyroidism relates to the post-operatory success rate, complications and operation time, pre-operatory imaging diagnosis is mostly helpful, also allowing a minimally invasive/selective procedure, thus playing a role in the modern era in the parathyroid domain since this minimal surgical act in addition to the pre-operatory localisation of the tumour provides similar cure rates to bilateral neck exploration [18,19,20]. Nevertheless, many parathyroid surgeons may currently perform 4-gland exploration routinely, precisely to avoid the shortcomings of preoperative localisation.

### Aim

We aimed to introduce the case of an adult female diagnosed with PHP displaying unexpected intra-operatory findings in relation to pre-operatory imaging modalities.

## 2. Method

This was a case report. The medical aspects were retrospectively registered. Biochemistry, hormonal and imaging panels were introduced, as well as intra-operatory (surgical) aspects. A brief literature overview is performed in the Discussion section with regard to potential pitfalls of pre-surgery imagery scans, particularly 99m-Tc scintigraphy in relation to pre-operatory concordant imaging results, and with its potential of differentiating between a thyroid and a parathyroid tissue.

## 3. Clinical Vignette

This was a non-smoking 56-year-old lady who was admitted after an accidental detection of increased serum calcium amid a routine evaluation that was conducted by her primary care physician. She had surgical menopause since the age of 40 for a benign uterine tumour and did not receive any hormone replacement therapy afterwards. Her family history was irrelevant; she has been known to have mild high blood pressure for the last decade (that was controlled under specific daily medication with beta blockers and calcium blocker—amlodipine). On admission, PHP was confirmed based on increased blood values of calcium and PTH (Table 1).

The patient had normal thyroid function and negative serum antibodies against thyroid. DXA (Dual-Energy X-ray Absorptiometry thorough a GE Lunar Prodigy device) confirmed osteoporosis (no prevalent vertebral fracture was detected at profile X-ray at the level of thoracic–lumbar spine) (Table 2).

A value of lumbar DXA-based TBS (trabecular bone score) of 1349 (iNsight) was considered a partially degraded microarchitecture (yet, close to the normal cut off of at least 1350) [21]. No kidney stone was detected at abdominal ultrasound, nor any component of multiple endocrine neoplasia syndromes. Noting the PHP–osteoporosis (of course, with an additional early-menopause-related component) and a value of serum total calcium above +1 mg/dL above the upper normal limit, the patient had an indication of parathyroidectomy [22]; thus, localisation scans were performed starting with neck ultrasound that suggested a right inferior parathyroid tumour (Figure 1).

99m-Tc parathyroid scintigraphy was performed (a dual technique with 99mTcpertechnetate and 99m-Tcsestamibi) and showed two areas of increased late uptake of 99m-Tc sestamibi at the level of the inferior pole of the left thyroid lobe and of the right thyroid lobe, respectively, suggesting two bilateral inferior parathyroid tumours (Figure 2).

These findings were confirmed at intravenous contrast CT scan (Figure 3).

Due to the imaging diagnosis of multi-glandular disease, other hormonal assays were carried out (such as calcitonin or metanephrines/normetanephrines) but were found negative for a diagnosis of multiple endocrine neoplasia. After performing parathyroid scintigraphy and a CT scan, neck ultrasound was re-performed and a potential mass of around 1 cm at the level of the left latero-cervical area was suspected as being a parathyroid tumour.

Herein, the patient was offered 5 mg of intravenous zoledronate in order to control hypercalcaemia (with a post-injection decrease in total calcium to 10.4 mg/dL) and to serve as an anti-osteoporotic regime (5 mg zoledronate per year) in addition to daily cholecalciferol of 1000 UI.

The patient was further referred to a one-time parathyroidectomy of both tumours. Firstly, the right inferior parathyroid tumour was removed, followed by an intra-operatory PTH assay that showed a very low value. Since the high-PTH-originating source seemed to be this adenoma, an extemporaneous exam was performed for the second mass (that was initially suspected to be a synchronous parathyroid adenoma, as well) and revealed an ectopic thyroid tissue. Thus, an intra-operatory decision was made for further removal with post-operatory histological confirmation of this ectopic thyroid tissue (with dilated follicles, and colloid content, without atypia or malignant elements). The right para-tracheal tumour was confirmed post operation as being a parathyroid adenoma without vascular invasion (Figure 4).

The post-surgery outcome was uneventful; the patient experienced a normalisation of calcium and PTH levels without any influence on thyroid function. The post-parathyroidectomy 1-month PTH was 64 pg/mL (Figure 5). The lady continued vitamin D replacement and a periodic check-up is required including annual DXA.

## 4. Discussion

We highlight some major aspects starting from this case in point: the potential pitfalls of pre-operatory imaging in PHP; the concordance/discordance of pre-parathyroidectomy localisation modalities; and the need of using an additional intra-operatory procedure in certain cases, which are clues for providing a distinction between pathological parathyroids and thyroid tissue.

### 4.1. Imaging Pitfalls in PHP

This case reveals an interesting and challenging aspect: incongruent findings between pre-operatory imaging captures pinpointing a double parathyroid adenoma in terms of 99m-Tc pertechnetate (TcO4)–sestamibi plan scintigraphy and a CT exam, on the one hand, and the intra/post-operatory confirmation with regard to one of these masses, one the other hand (one of the presumably “parathyroid” tumours proved to be an ectopic thyroid tissue with no malignancy features). An intra-operatory decision was mandatory amid lowering PTH values after the resection of the first mass. Since Tc scintigraphy was found positive on both sides, a potential non-functional parathyroid tumour was suspected on the left (thus explaining the PTH normalisation), while an ectopic thyroid tissue seemed more likely based on its macroscopic features (and this was finally confirmed). The removal of this second mass was based on the fact that it was not a normal parathyroid, nor a thyroid gland; hence, it represented an individual decision in this lady’s case, not a matter of guidance. However, in addition to the results upon the first-hand histological (extemporaneous) report, a consultation with her current endocrinologist was performed amidst surgery in order to proceed with the second tumour removal.

Generally, a good localisation study before performing parathyroidectomy is helpful for the surgeon and avoids unnecessary prolonged surgery time or even a redo of parathyroidectomy, thus allowing a minimally invasive approach (and a shorter hospitalisation stay and increased patient’ comfort) [12,13]. Also, an adequate pre-operatory localisation, regardless of the methods, improves the PTH values within the first minutes after tumour removal (as seen in this mentioned vignette) [23].

Generally, the use of the 99m-Tc sestamibi-based scan is traditionally a part of the pre-parathyroidectomy preparation panel (740–924 MBq). A 99m-Tc radiotracer may be administered via injection (intravenously) or orally and it may also be used for thyroid scintigraphy (sodium pertechnetate), both for adults (1–10 mCi) and children (60–80 µCi/kg). Increased homogeneous tracer uptake is suggestive for thyroid/parathyroid tissue in dual scintigraphy (sestamibi and pertechnetate), mostly depending on the uptake timing (early for thyroid and late/delayed for parathyroid at the moment when thyroid images are already washed out) [24].

However, despite performing imaging techniques, some pitfalls cannot be predicted, especially if concordant localisation results are found with different methods (for instance, in our case, CT combined with 99m-Tc plan scintigraphy); thus, the appeal of using an additional technique did not seem justified [12,13].

A spectrum of bias relates to the clinical, hormonal and anatomical aspects in PHP as well as technical issues, including their availability/access in one centre (and associated experience of the medical team). Inter-observer differences have been reported with concern to Tc-99m-based scintigraphy, but, generally, ultrasound is regarded as the most commonly recognised subject of pitfalls [25]. A higher rate of localisation failure was also described in ectopic intra-thyroid parathyroid adenomas [26]. Also, a lower PTH value at baseline is correlated with a negative finding at 99m-Tc scintigraphy; thus, in these cases a combination of several modalities is required from the start [27]. A multi-glandular disease (generally representing 20% of the adult cases with PHP, particularly those with a strong genetic background) associates with a higher rate of identification failure regardless of the pre-surgery functional imaging modalities (particularly when ectopic pathological glands are also involved) [28]. 99m-Tc sestamibi plan scintigraphy has been used not only in PHP, but also in secondary (chronic kidney disease-associated) hyperparathyroidism, with the sensitivity for paediatric cases being lower than seen in the adult population (for example, 40% versus 70%) with an increased rate of reduced radiotracer uptake at the thyroid level (42% versus 2%) [29]. The pre-surgery detection rate is generally higher in PHP than in the secondary type [30].

Another dual-phase scintigraphy, namely dual tracer (99m-Tc and 123 Iodine), provides simultaneous images that avoid the subtraction-related artefacts and might prove beneficial in multi-glandular parathyroid disease in addition to thyroid gland-related information [31,32]. The traditional issues of blocked thyroid uptake after recent iodine exposure (as seen after using iodine contrast CT) or in individuals under chronic levothyroxine replacement therapy should be taken into consideration when using iodine scintigraphy [33]. Of course, there is a standard issue of availability; for instance, at the moment when we evaluated the patient, iodine-based scintigraphy was not available.

In order to enhance the performance of 99m-Tc dual-phase plan scintigraphy, an alternative is represented by 99m-Tc-MIBI SPECT/CT fusion imaging [34]. SPECT/CT offers an increased sensitivity and accuracy for location diagnosis and it became a first-line option in some centres [35] (but not in ours) or it may be applied as an elegant alternative in difficult cases such as suspected recurrent tumours or carcinomas [36]. One limit was found to be a reduced pathological tumour weight (false negative results) [37] and some data showed a decreased rate of localisation in normocalcaemic PHP versus PHP, associating a classical presentation with high levels of serum calcium that might involve a smaller parathyroid gland. On the contrary, a hypercalcaemia-related inhibitory effect of the radiotracer uptake has been reported, too [12,13,38]. Also, the use of calcium blockers (as in our case according to patient’s medical records in order to control the arterial hypertension) and calcimimetics such as cinacalcet may reduce the 99m-Tc MIBI uptake [12,13] (Figure 6).

### 4.2. Concordance of Pre-Operatory Localisation Studies in PHP

This was a case of adult PHP, whereas triple localisation methods were used before parathyroidectomy, showing concordant results; however, the second parathyroid adenoma was a false positive image. In addition to 99m-Tc-based assessment, a standard CT scan was conducted in this mentioned situation. Despite recent progress in the era of imaging in PHP, CT remains a mostly used and feasible approach as part of a routine exam in real-life medicine [39,40]. Dual-phase (non-enhanced and arterial) CT protocols increase the accuracy of results, in both the adult and paediatric (but less accurate) population with PHP [41]. Of note, CT should be avoided if possible in children, especially as non-radiating techniques are available. Alternatively, dual-energy CT (DECT) showed similar accuracy with conventional imaging techniques [42]. Notably, the concordance of pre-operatory localisation diagnosis did not raise the suspicion of further using an additional imaging technique in this instance.

Some data suggested that PTH levels that are not extremely elevated might mislead the interpretation of 99m-Tc-based scintigraphy. For example, one study correlated the cut off values of serum calcium and PTH with positive Tc-99m-MIBI (methoxyisobutylisonitrile) parathyroid scintigraphy (and positive parathyroid subtraction) in patients diagnosed with PHP (median age of 60, between 22 and 78 years). PTH (not serum calcium) statistically significantly correlated with 99m-Tc scintigraphy findings [43]. This implies that cases with a PTH level around 100 pg/mL (as seen in our case) should be taken into consideration in interpreting scintigraphy.

Recently, 11C-methionine, 11C-choline or 18F-fluorocholine PET/CT showed encouraging outcomes, especially in patients who underwent unsuccessful parathyroidectomy or had negative or discordant imaging results pre-surgery, with these patients being alternatively candidates to SPECT/CT, as well [44,45,46,47]. Head-to-head studies (versus 99m-Tc sestamibi SPECT/CT) showed comparable results despite using a different tracer for parathyroid glands [48] or even an improvement in detection rate, for instance, from 88% for 99m-Tc-sestamibi SPECT/CT to 98% according to one recent study [49] or a sensitivity of 99% for 18F-fluorocholine PET/CT versus 75% for ultrasound versus 65% for 99m-Tc sestamibi scintigraphy versus 89.9% for ultrasound combined with this type of scintigraphy according to another cohort from 2022 [50].

As mentioned, a lower detection rate was described in multi-glandular disease/hyperplasia versus single-gland involvement [48,49]. Alternatively, 18F-fluorocholine PET/ultrasound fusion imaging might bring supplementary benefits [51]. 11C-choline PET/CT might help in cases with negative/discordant data when using traditional methods such as CT and/or Tc scintigraphy, with approximately 93% of such cases being true positive [52]. 18F-fluorocholine PET/MRI has recently been proven to have a similar or higher accuracy than 99m-Tc sestamibi scintigraphy, with the association of these two modalities being currently recommended [53]. Currently, there is no such thing as a unique algorithm of pre-operatory functional multimodal imaging and the use of PET/CT or PET/MRI in PHP is not standardised yet, largely depending on their availably in one centre and local protocols [54].

An alternative to the mentioned pre-operatory imaging methods was proposed to be ultrasound-guided parathyroid fine-needle aspiration (with PTH washout), but so far, there are not so many large studies to address this issue, which is not conventionally approved in many centres [55]. However, one study from 2023 showed a similar positive predictive value with 99m-Tc sestamibi scintigraphy; thus, it might become a first-line imaging modality in a selective subgroup of subjects with PHP [56]. Also, the rate of non-localisation in patients who consequently required a redo of parathyroidectomy is higher; thus, parathyroid vein sampling was proposed as an intra-operatory additional method for those subjects with a negative pre-operatory imaging diagnosis in recurrent/persistent PHP [57]. Of note, intra-operatory PTH monitoring or assessments (as we used in our case, too) represent an essential clue for the parathyroidectomy success [58,59].

Another approach involves an alternative to radiologist-based—or endocrinologist-based—pre-operatory neck ultrasound, namely a similar pre-incision ultrasound that is carried out by the surgeon on the operating table while the patient is under general anaesthesia just before the actual parathyroid removal. This intra-operatory “before skin incision” localisation procedure might enhance the results of pre-operatory echography, thus allowing a minimally invasive parathyroidectomy. However, the implementation of such a protocol largely depends on each centre standards rather than being a guideline recommendation [60,61,62,63]. Overall, negative or discordant pre-operatory imaging results in PHP do not exclude a successful parathyroidectomy in the hands of a skilled surgeon [44,45].

### 4.3. Ectopic Thyroid Tissue

While false negative results at 99m-Tc sestamibi scintigraphy may involve small parathyroid tumours or parathyroid cysts, false positive results relate to abnormal tracer uptake at the thyroid level (in our case, only at ectopic tissue, not at normal thyroid gland). The fact that 99m-Tc MIBI is up-taken in highly metabolic lesions might imply a different cellular turnover in ectopic, not orthotopic, thyroids, as has similarly been reported (false positive results) in different types of head and neck cancers, even breast and lung neoplasia [64,65,66,67]. Hence, in this case, the incidental detection of the ectopic thyroid was globally due to the first diagnosis of PHP and associated management, while its actual recognition started during surgery from an unexpected PTH drop after the excision of the first tumour. We identified a similarity with another case published in 2019: this was a 52-year-old male confirmed with PHP. Pre-operatory MIBI SPECT/CT confirmed a late uptake on two focal areas at the level of the left thyroid lobe, but the post-surgery histological report showed that only one was a parathyroid adenoma and the other was a thyroid hyperplasia. Notably, pre-operatory 99m-Tc MIBI planar scintigraphy was not relevant, only showing positive at SPECT/CT; thus, we may conclude that false positive images due to synchronous concurrent thyroid conditions may be detected even with regard to advanced imaging modalities [68].

On the other hand, with concern to parathyroid tumours, pre-operatory false negative results involve between 5.7 and 25% of the patients who perform 99m-Tc sestamibi SPECT scintigraphy, and they are related to the limits of resolution of the technique or a prolonged time from scintigraphy to surgery [69,70]. Also, in patients with large goitres, there is a higher rate of false negative results at ultrasound and 99m-Tc sestamibi scintigraphy [71]. As mentioned, a negative parathyroid localisation result (even caused by thyroid issues) might be overcome by an experienced surgeon in the field of parathyroid tumours, so-called “the sestamibi paradox” [72].

A similar case regarding both parathyroid and thyroid pathological findings was reported in 2023 on a subject who underwent 99mTc-sestamibi scintigraphy: the lack of orthotopic uptake at the level of the thyroid area was associated with an ectopic thyroid tissue that was identified at the lingual level, while, synchronously, there was an ectopic mediastinal parathyroid tumour [73]. Another very rare scenario involves the concomitant diagnosis of PHP and giant-goitre-associated thyrotoxicosis; in this situation, hypercalcaemia-related hyperthyroidism may mask the biological recognition of PHP and 99m-Tc sestamibi scintigraphy might be found as false negative at baseline [74]. Also, 99m-Tc scintigraphy has been used for thyroid hemi-agenesis, showing an increased unilateral uptake amid this interesting developmental disease of the gland [75,76].

Finally, some potential implications may be related to using a dual tracer at 99m-Tc scintigraphy. The 99m-Tc pertechnetate radiotracer may be up-taken by the thyroid and other organs such as the gastric mucosa [77]. Non-iodine-based methods for the functional and anatomic study of the thyroid gland include not only 99m-Tc sestamibi scintigraphy, but also 18F-FDG (^18^F-fluoro-2-deoxy-d-glucose) PET/CT [78]. Further on, 99m-Tc sestamibi has been applied (via different quantitative parameters) in order to stratify the malignancy risk of the cold nodules (that do not uptake 99m-Tc TcO4) with indeterminate results following a thyroid-fine-needle-aspiration-associated cytological exam [79,80,81]. However, in our case, the late uptake amid dual-tracer plan scintigraphy was registered only at the level of ectopic thyroid tissue synchronously with the parathyroid adenoma, and not at the physiological thyroid. Hybrid imaging modalities are most probably required in both thyroid and parathyroid anomalies, but, essentially, awareness of the potential pitfalls is mandatory from the clinical and surgical perspectives [82].

## 5. Conclusions

This was a case of adult PHP, whereas triple localisation methods were used before parathyroidectomy, showing concordant results. However, the second parathyroid adenoma was a false positive image and an ectopic thyroid tissue was confirmed. The pre-operatory index of suspicion was non-existent in this patient. Thus, we may conclude that in a selected subgroup of individuals, hybrid imaging modalities might prove useful if both thyroid and parathyroid conditions are present, but, essentially, awareness of such potential pitfalls is mandatory from the endocrine and surgical perspectives. Current gaps in imaging knowledge to guide us in these specific areas are expected to be solved by the significant progress in functional imaging modalities. However, the act of surgery, including the decision of a PTH assay or extemporaneous exam (as seen in our case), represents the major key to a successful removal procedure.

## Figures and Tables

**Figure 1 medicina-60-00015-f001:**
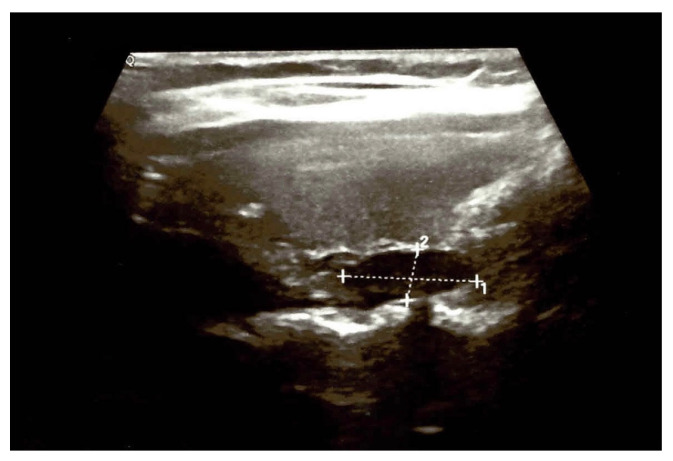
Neck ultrasound showing a right thyroid lobe of 2/2/5 cm; a left thyroid lobe of 2/1/4 cm with hypoechoic, inhomogeneous pattern; and, posterior and inferior to the right thyroid lobe, a hypoechoic nodule, inhomogeneous, with no vascularisation, of 1.56/0.6/0.7 cm, suggestive for a parathyroid adenoma.

**Figure 2 medicina-60-00015-f002:**
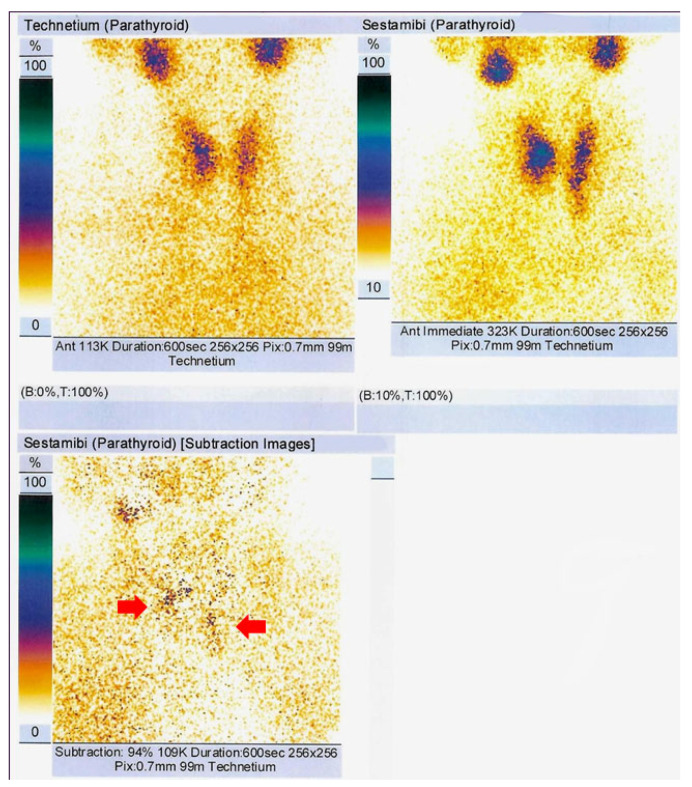
99m-Tc parathyroid scintigraphy with 99m-Tc pertechnetate (185 MBq) and 99m-Tc sestamibi (740 mBq; effective dose of 9.06 mSv); 94% subtraction captures (at 60 min show two late-uptake areas at the level of left and right inferior thyroid lobes, suggestive for parathyroid adenomas (red arrow).

**Figure 3 medicina-60-00015-f003:**
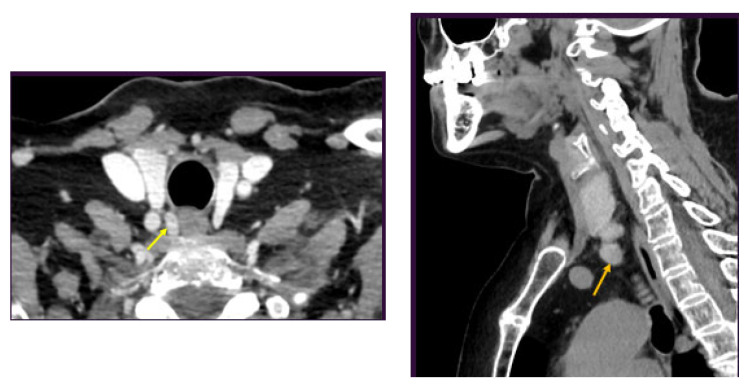
CT scan showing an oval-to-round, well-circumscribed, iodophile, slightly heterogeneous nodule at the right para-tracheal level (in the superior mediastinum–manubrium level) of 1.12/1.33/1.40 cm (yellow arrow; transversal plan), respectively, and an oval, well-circumscribed, iodophile nodule at latero-cervical, left para-oesophageal, clavicular level, of 0.56/1.27/1.72 cm (orange arrow; sagittal plan).

**Figure 4 medicina-60-00015-f004:**
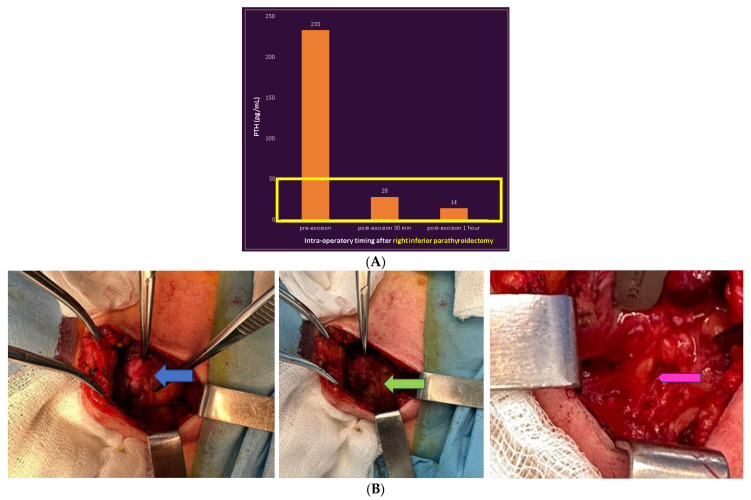
Surgical aspects: (**A**) PTH assays after right inferior parathyroid tumour removal. (**B**,**C**) Intra-operatory captures: (**B**) right inferior parathyroid tumour (blue arrow); right inferior thyroid pedicle (green arrow); right recurrent laryngeal nerve (pink arrow). (**C**) Ectopic (upper mediastinal) thyroid tissue that, pre-operation, mimicked an additional left parathyroid tumour (white arrow); left recurrent laryngeal nerve (pink arrow). (**D**) Post-operatory specimen: macroscopic aspect of the right inferior parathyroid adenoma.

**Figure 5 medicina-60-00015-f005:**
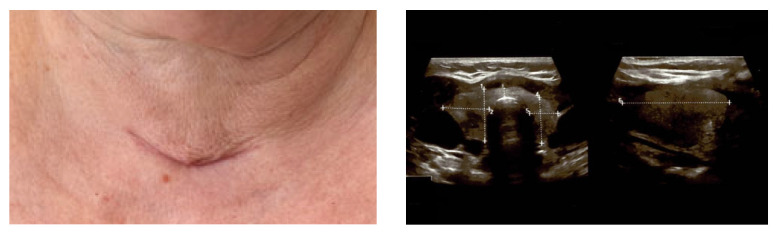
Post-parathyroidectomy scar within the first few weeks (**left**); anterior neck ultrasound showing thyroid features similar to pre-operatory findings and no remnants at the level, whereas both masses have been removed (**right**).

**Figure 6 medicina-60-00015-f006:**
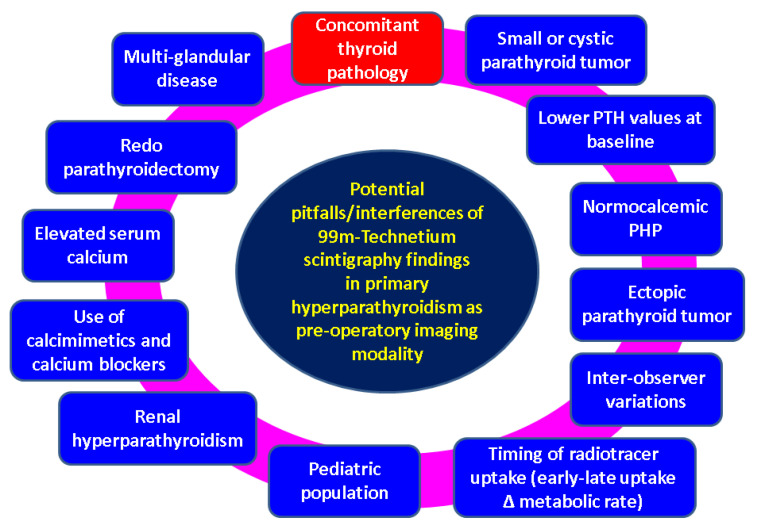
Sneak peak of potential pitfalls when addressing the results of 99m-Tc sestamibi scintigraphy in PHP [1,2,3,4,5,6,7,8,9,10,11,12,13,14,15,16,17,18,19,20,21,22,23,24,25,26,27,28,29,30,31,32,33,34,35,36,37,38,39,40,41,42,43,44,45,46,47,48,49,50,51,52,53,54,55,56,57,58,59,60,61,62,63,64,65,66,67,68,69,70,71,72,73,74,75,76,77,78,79,80].

**Table 1 medicina-60-00015-t001:** Biochemistry and hormonal level on first admission for PHP on a 56-year-old female (* bone formation markers; ** bone resorption markers).

Parameter	Patient’s Value	Normal Range
Total serum calcium (mg/dL)	11.3	8.5–10.2
Ionised calcium (mg/dL)	5.07	3.9–4.9
Total proteins(g/dL)	6.8	6.5–8.7
Phosphorus (mg/dL)	2.8	2.5–4.5
Creatinine (mg/dL)	0.77	0.5–1.1
Urea (mg/dL)	35	15–50
Hormones of mineral metabolism		
25-hydroxyvitamin D (ng/mL)	34	20–100
PTH (pg/mL)	103.1	15–65
Bone turnover markers		
Osteocalcin (ng/mL) *	48.2	15–46
Alkaline phosphatase(U/L) *	117	38–105
P1NP (ng/mL) *	103.1	14.28–58.92
CrossLaps (ng/mL) **	1.05	0.33–0.782

**Table 2 medicina-60-00015-t002:** Central DXA (GE Lunar Prodigy) report in a menopausal female with PHP with osteoporosis confirmation based on T-score.

Region	Bone Mineral Density (g/cm^2^)	T-Score (SD)	Z-Score (SD)
Lumbar spine L1–4	0.764	−3.5	−2.5
Femoral neck	0.640	−2.8	−1.9
Total hip	0.606	−3.3	−2.6
1/3 distal radius	0.592	−3.2	−2.7

## Data Availability

The original data generated and analysed for this case presentation are included in the published article.

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
