# Peer review of "Pitfalls of DualTracer 99m-Technetium (Tc) Pertechnetate and Sestamibi Scintigraphy before Parathyroidectomy: Between Primary-Hyperparathyroidism-Associated Parathyroid Tumour and Ectopic Thyroid Tissue"

_medicina, 2023, doi:10.3390/medicina60010015_

Round 1

Reviewer 1 Report

Comments and Suggestions for Authors

Dear Authors,

This study presents a very interesting case Pitfalls of dual tracer 99m-Techetium (Tc) pertechnetate andsestamibi scintigraphy before parathyroidectomy: between primary hyperparathyroidism – associated parathyroid tumor and ectopic thyroid tissue.  The authors made a well structured presentation of used method and data presentation, as well as the detailed discussion on each question of importance. I consider your work an interesting contribution of the topic that should be accepted for the publication. Please add a comment on the usefulness of Choline  C-11 PET.

Author Response

Response to Review 1 Comments

Dear Reviewer,

Thank you very much for your time and your effort to review our manuscript.

We are very grateful for providing your valuable feedback on the article.

Here is our response and related amendment that has been made in the manuscript according to your review (marked in yellow color).

Dear Authors,

This study presents a very interesting case Pitfalls of dual tracer 99m-Techetium (Tc) pertechnetate and sestamibi scintigraphy before parathyroidectomy: between primary hyperparathyroidism – associated parathyroid tumor and ectopic thyroid tissue.  The authors made a well-structured presentation of used method and data presentation, as well as the detailed discussion on each question of importance. I consider your work an interesting contribution of the topic that should be accepted for the publication.

Thank you very much. We really appreciate it!

Please add a comment on the usefulness of Choline  C-11 PET.

Thank you very much. The following data has been provided with respect to C-Choline PET scan:

“Further on, PET/CT (positron emission tomography), for instance, with 18F-Fluorocholine or 11C-methionine and PET/MRI (which requires less radiation dose than PET/CT or CT scan) provides useful dynamic captures [7,14,15].”

“Recently, 11C-methionine, 11C-choline or 18F-fluorocholine PET/CT showed encouraging outcomes, especially in patients who underwent unsuccessful parathyroidectomy or had pre-surgery negative or discordant imaging results, these patients being alternatively candidates to SPECT/CT, as well”

“11C-choline PET/CT might help in cases with negative/discordant data when using traditional methods such as CT and/or Tc scintigraphy, approximately 93% of such cases being true positive [52].”

Thank you very much.

Reviewer 2 Report

Comments and Suggestions for Authors This is a case report addressing a subject whose preoperative findings by two acknowledged imaging techniques did not match the intraoperative findings by the surgeon in situ. It addresses the question of the necessity of additional imaging methods even in cases when standard preoperative procedures give concordant results and leave no doubt for diagnosis.
Hence, I find the topic relevant for the endocrinologist and surgical society. To the best of my knowledge, no similar reports have been published. Apart from minor grammatical errors throughout the paper, the report is detailed, well written, and gives a thorough overview of the case. As for improvements, the tables and figures are sufficiently informative while the the Introduction section provides enough information to follow the narrative. The conclusions are also sound, providing an explanation for the described report. Hence, I see no reason why the manuscript could not be accepted in its present form, after minor English editing. Comments on the Quality of English Language

Minor English editing required

Author Response

Response to Review 2 Comments

Dear Reviewer,

Thank you very much for your time and your effort to review our manuscript.

We are very grateful for your insightful comments and observations, also, for providing your valuable feedback on the article.

Here is a point-by-point response and related amendments that have been made in the manuscript according to your review (marked in yellow color).

This is a case report addressing a subject whose preoperative findings by two acknowledged imaging techniques did not match the intraoperative findings by the surgeon in situ. It addresses the question of the necessity of additional imaging methods even in cases when standard preoperative procedures give concordant results and leave no doubt for diagnosis.Hence, I find the topic relevant for the endocrinologist and surgical society. To the best of my knowledge, no similar reports have been published. Apart from minor grammatical errors throughout the paper, the report is detailed, well written, and gives a thorough overview of the case. As for improvements, the tables and figures are sufficiently informative while the Introduction section provides enough information to follow the narrative. The conclusions are also sound, providing an explanation for the described report. Hence, I see no reason why the manuscript could not be accepted in its present form, after minor English editing.

Thank you very much. We really appreciate it!

Comments on the Quality of English Language: Minor English editing required.

Thank you very much. We re-edited the paper. Thank you

Reviewer 3 Report

Comments and Suggestions for Authors

This is a case report describing a false positive finding on preoperative imaging of bilateral parathyroids reported on sestamibi scan in a woman with primary hyperparathyroidism, with intraoperative findings confirming single gland disease. The authors describe the weaknesses of sestamibi scans and state the need for intraoperative adjuncts such as intraoperative parathyroid hormone monitoring.

It is well known that preoperative parathyroid localization techniques each have shortcomings. I have the following specific comments:

1.     Introduction and abstract. It is mentioned that preoperative imaging is required to localize abnormal parathyroid tissue prior to parathyroidectomy. However, it is not a necessity. Many parathyroid surgeons perform 4-gland exploration routinely, precisely to avoid the shortcomings of preoperative localization.

2.     Ultrasound is describe as a type of “radionuclide imaging”, whereas it is not.

3.     The use of selective venous sampling is not simply another technique to be used alongside other radiological modalities, but an invasive tool that is largely reserved for re-operative cases in which imaging has not identified a target.

4.     Figure 3 describes a “left” lesion, whereas the arrow is pointing to a lesion on the right side.

5.     As intraoperative parathyroid monitoring was utilized and a significant (and adequate) decline in PTH was seen after resection of the right sided lesion, why was further exploration carried out?

6.     The preoperative imaging in this case was not concordant – a single, right sided lesion, was identified on ultrasound, whereas the sestamibi (and possibly CT) reported bilateral lesions.

7.     CT should be avoided if possible in children, especially as non-radiating techniques are available.

Comments on the Quality of English Language

The quality is poor and many sentences are not written comprehensively. 

Author Response

Response to Review 3 Comments

Dear Reviewer,

Thank you very much for your time and your effort to review our manuscript.

We are very grateful for your insightful comments and observations, also, for providing your valuable feedback on the article.

Here is a point-by-point response and related amendments that have been made in the manuscript according to your review (marked in yellow color).

This is a case report describing a false positive finding on preoperative imaging of bilateral parathyroids reported on sestamibi scan in a woman with primary hyperparathyroidism, with intraoperative findings confirming single gland disease. The authors describe the weaknesses of sestamibi scans and state the need for intraoperative adjuncts such as intraoperative parathyroid hormone monitoring.

It is well known that preoperative parathyroid localization techniques each have shortcomings. I have the following specific comments:

Introduction and abstract. It is mentioned that preoperative imaging is required to localize abnormal parathyroid tissue prior to parathyroidectomy. However, it is not a necessity. Many parathyroid surgeons perform 4-gland exploration routinely, precisely to avoid the shortcomings of preoperative localization.

Thank you very much. We followed your recommendation and introduced this specification in Abstract and Introduction sections.

Ultrasound is described as a type of “radionuclide imaging”, whereas it is not.

Thank you very much. We corrected the sentence as following: “Parathyroid imaging represents a mandatory pre-operatory step and currently a large spectrum of tools are available starting with the mostly frequently used, namely cervical ultrasound.” Thank you

The use of selective venous sampling is not simply another technique to be used alongside other radiological modalities, but an invasive tool that is largely reserved for re-operative cases in which imaging has not identified a target.

Thank you very much. We followed your recommendation and introduced this mentioned specification at Introduction.

Figure 3 describes a “left” lesion, whereas the arrow is pointing to a lesion on the right side.

Thank you very much. We corrected it.

As intraoperative parathyroid monitoring was utilized and a significant (and adequate) decline in PTH was seen after resection of the right sided lesion, why was further exploration carried out?

Thank you very much for this interesting observation. We explained at Discussion this decision. “Since Tc scintigraphy was found positive on both sides, a potential non-functional parathyroid tumour has been suspected on the left (thus explaining the PTH normalization) while an ectopic thyroid tissue seemed more likely based on its macroscopic features (and this was finally confirmed). The removal of this second mass was based on the fact that it was not a normal parathyroid, nor thyroid gland; hence it represented an individual decision in this lady case, not a matter of guideline. However, in addition to the results upon first hand histological (extemporaneous) report, a consultation with her current endocrinologist was done amidst surgery in order to carry out with the second tumour removal.” Thank you   

 The preoperative imaging in this case was not concordant – a single, right sided lesion, was identified on ultrasound, whereas the sestamibi (and possibly CT) reported bilateral lesions.

Thank you very much. After sestamibi and CT scan, ultrasound was re-done and suggested a mass of 1 cm on the left as being a potential parathyroid tumor. We introduced this specification. Thank you

CT should be avoided if possible in children, especially as non-radiating techniques are available.

Thank you very much. We followed your recommendation and introduced this mentioned specification at Discussion. Thank you

Comments on the Quality of English Language. The quality is poor and many sentences are not written comprehensively. 

Thank you very much. We re-edited the paper. Thank you

Thank you very much
